# Effects of *Nannochloropsis salina* Fermented Oil on Proliferation of Human Dermal Papilla Cells and Hair Growth

**DOI:** 10.3390/ijms25158231

**Published:** 2024-07-28

**Authors:** Ming Ying, Jialin Zhou, Zuye Zeng, Shuangfei Li, Xuewei Yang

**Affiliations:** Guangdong Technology Research Center for Marine Algal Bioengineering, Guangdong Key Laboratory of Plant Epigenetics, College of Life Sciences and Oceanography, Shenzhen University, Shenzhen 518060, China; yingming@szu.edu.cn (M.Y.); 2100251023@email.szu.edu.cn (J.Z.); 2100251019@email.szu.edu.cn (Z.Z.); sfli@szu.edu.cn (S.L.)

**Keywords:** alopecia, *Nannochloropsis salina* fermented oil, eicosapentaenoic acid, human dermal papilla cells, antioxidant capacity

## Abstract

The hair follicle is the basis of hair regeneration, and the dermal papilla is one of the most important structures in hair regeneration. New intervention and reversal strategies for hair loss may arise due to the prevention of oxidative stress. GC/MS analysis was used to determine the compounds contained in NSO. Then, NSO was applied to DPC for cell proliferation and oxidative stress experiments. RNA-seq was performed in cells treated with NSO and minoxidil. The quantitative real-time polymerase chain reaction (qRT-PCR) was applied to verify the gene expression. The effects of NSO on hair length, weight, the number and depth of hair follicles, and the dermal thickness were also studied. GC/MS analysis showed that the main components of NSO were eicosapentaenoic acid, palmitic acid, and linoleic acid. NSO promotes DPC proliferation and reduces H_2_O_2_-mediated oxidative damage. NSO can also activate hair growth-related pathways and upregulate antioxidant-related genes analyzed by gene profiling. The topical application of NSO significantly promotes hair growth and increases hair length and weight in mice. NSO extract promotes hair growth and effectively inhibits oxidative stress, which is beneficial for the prevention and treatment of hair loss.

## 1. Introduction

Hair loss is a widespread and painful disease that affects most middle-aged men and women around the world, with significant psychological and social impacts on patients. Compared to the general population, not only are mental illnesses, such as depression and anxiety, extremely common among patients who have lost their hair, but these individuals also show a loss of confidence [1].

Hair follicle irregularities and atrophy are the hallmarks of hair loss, a chronic illness whose periodic growth is regulated by a variety of circumstances [2]. The hair follicle is the basis of hair regeneration, and the dermal papilla is one of the most critical structures in hair regeneration [3]. Dermal papilla cells (DPCs) can form signal molecular networks by secreting a variety of bioactive substances such as vascular endothelial growth factor (VEGF) that is responsible for cell proliferation and differentiation in the hair follicle. Many cytokines, receptors, and active proteins play their biological roles, such as leading to hair growth and development, cycle regulation, and the maintenance of hair growth [4]. Hair growth is determined by periodic changes in the hair follicles, including anagen (in an active growth state, about 90% of hairs), catagen (in a state of degeneration, less than 10% of hairs), and telogen (in the condition of rest, 5% to 10% of hairs) [5,6]. Anagen refers to the stage of active hair growth during which pigmented hair shafts are produced and hair follicles reach their maximum length and volume. The rest period starts when the hair follicles go into telogen, a generally calm phase. This immobile or inactive stage of the hair development cycle is characterized by a marked reduction in hair follicle proliferation and metabolic activity [5]. When a large amount of growing hair is stimulated and stops growing, it will enter catagen and telogen, causing alopecia [6]. 

Numerous variables, including metabolic syndrome, UV radiation, and cigarette smoking, are linked to oxidative stress in cells and, consequently, hair loss [7]. An increasing amount of evidence indicates the impact of oxidative stress on hair loss, and preventing oxidative stress may provide new strategies for intervening and reversing hair loss and even hair graying [8]. Research has verified that persons with alopecia areata undergo elevated levels of oxidative stress in contrast to those in good health [9]. Moreover, studies employing mice as a model have demonstrated that H_2_O_2_-induced oxidative stress is the primary cause of hair loss brought on by hair dyes [10]. Episodes of androgenetic alopecia (AGA) are caused by oxidative stress irritating tumor growth factor-*β*, a recognized inhibitor of hair follicles in DPCs [11]. Research has demonstrated that reducing oxidative stress is one of the key strategies for overcoming androgen- or oxidative stress-induced hair loss. Troxerutin, for instance, prevented DPC damage brought on by oxidative stress [12].

Currently, the Food and Drug Administration of the USA has approved only two medications to treat androgenetic alopecia, the most prevalent form of hair loss: finasteride and minoxidil [2]. Nevertheless, both substances have a high potential for recurrence upon withdrawal and have adverse effects that include skin allergies and hirsutism of the hands and face [13,14]. Furthermore, research has revealed that a variety of growth factors included in platelet-rich plasma (PRP) contribute to cell division and proliferation during the formation of hair. Injecting PRP can improve the hair density and straight diameter of female patients with androgenetic alopecia; although there will be some adverse reactions, such as headache, redness swelling, and bleeding after injection, it is still a safe and effective method [15,16]. Claristone can also be used to treat androgen-dependent skin diseases, including androgenetic alopecia. However, the research evidence on this drug in China is not sufficient, and more clinical studies are needed to verify it [17]. Moreover, although hair transplantation has a certain therapeutic effect, its clinical application is limited due to its large trauma, high cost, and multiple operations. The effect of acupuncture and other non-drug treatments is not ideal. Consequently, it is essential to conduct research and create novel therapies that can both slow down hair loss and promote hair growth. Therefore, these efforts should pay attention especially to natural substances, which could lead to less severe reactions. The investigation of natural products that promote the growth of dermal papilla cells mostly focuses on plant extracts. The ginsenoside Rg4 in natural ginseng activates hair growth-related pathways in DPCs, such as the Akt/GSK3*β*/*β*-catenin signaling pathway that promotes the hair induction properties of DPCs, indicating that Rg4 may be beneficial for hair growth [18]. Through decreasing the mRNA expression of *TGF-β1* while increasing the expression of *VEGF* and *IGF-1*, the *Nelumbinis Semen* (NS) extract contributes to the prevention of hair loss [19]. 

Fatty acids known as polyunsaturated fatty acids (PUFAs) include long-chain fatty acids, with two or more double bonds on the carbon backbone. Linoleic acid (LA) separated from *Malva verticillata* seeds could encourage the proliferation of DPCs and hair development by accelerating the cell cycle and growth factor secretion through activating Wnt/*β*-catenin signaling. Arachidonic acid (AA) and eicosapentaenoic acid (EPA) are omega-6 and omega-3 fatty acids, respectively, which can greatly affect the membrane phospholipids or signaling molecule creation such as eicosanoids. AA could improve the expression of hair growth-related factors like *HGF*, *FGF-7*, and *FGF-10* and phosphorylate the transcription factors ERK, CREB, and Akt, together with increasing the expression level of bcl-2 to enhance the activity of human dermal papilla cells (DPCs) [20]. The oil content of *Nannochloropsis salina* is as high as 37%~60%, especially the content of eicosapentaenoic acid (EPA), which has the effect of reducing blood lipids and preventing cardiovascular disease [21]. Compared with other sources, *Nannochloropsis salina* grows fast, has high oil content, and does not occupy cultivated land. The fatty acid composition is stable, the culture and reproduction are simple, and the growth cycle is short [22,23]. As a result, *Nannochloropsis salina* has enormous development potential as a source of a variety of omega-3 unsaturated fatty acids and is widely employed in aquaculture [24,25], biodiesel preparation [26], and the manufacturing of high value-added human nutritious dietary supplements [27]. However, there are few studies on the application of microalgae extracts to hair loss presently, mainly focusing on process optimization [28], agricultural production [29,30], antioxidant and anticancer effects [31,32], etc.

Currently, the only effective drugs on the market for treating hair loss are minoxidil and finasteride, which are prone to relapse after stopping and can cause a series of side effects such as skin allergies. Therefore, in order to develop a new type of therapeutic drug that is natural, safe, and fast-acting, this study discussed the effects of Nannochloropsis salina fermented oil extract on human dermal papilla cells and model mice hair growth, evaluated its role in promoting hair growth, clarified its mechanism, and provided valuable theoretical references for the further study of Nannochloropsis salina fermented oil in the prevention and treatment of hair loss.

## 2. Results

### 2.1. The Effect of NSO on the Vitality of DPCs with H_2_O_2_-Induced Damage

For the mechanisms involved in hair growth and the hair cycle, the proliferation of DPCs is essential [33]. To investigate whether NSO has a proliferative effect on DPCs, DPCs were cultured with various concentrations of NSO for 12 h, 36 h, and 60 h. The results showed that the cell proliferation was greatly enhanced by NSO during various stages of cultivation (12 h, 36 h, 60 h). Compared with the minoxidil group, NSO significantly stimulated the cell growth by 1.05–4.30 times, 1.06–2.02 times, and 0.80–2.59 times in 12 h, 36 h, 60 h, respectively (Figure 1a–c). Notably, the positive effect of NSO on cell proliferation (increased up to 4.30 times) was observed since the initiation stage (12 h), indicating that compared with the minoxidil group, NSO enhanced cell growth in the earlier period.

Oxidative damage is an important factor causing hair loss, so it is very important to study NSO for the repair of oxygen-damaged cells [10]. The results showed that in comparison to the positive group, the NSO treatment group was able to effectively improve the cell survival rate by 0.99–3.29 times, 1.08–2.38 times, and 1.34–8.99 times at 12 h, 36 h, and 60 h, respectively (Figure 2a–c). This indicated that the longer the cultivation time of NSO, the better its resistance to oxidative stress. These results suggested that pretreatment with NSO increased the ability against H_2_O_2_-mediated cytotoxicity in DPCs. 

### 2.2. The Effect of NSO on the H_2_O_2_-Induced ROS Production

To investigate the antioxidant properties of NSO, DCFH-DA was used as a fluorescence probe to measure the ability of NSO to protect DPCs from ROS effects caused by H_2_O_2_ [34]. After 1.5 h of treatment, in comparison with the control group, H_2_O_2_-treated cells showed a greater amount of mean fluorescence intensity (26.33 ± 2.29), but if cells were pretreated with NSO for 12 h, the increase in ROS content was prevented, as seen in the mean fluorescence intensity of the 10 μg/mL and 20 μg/mL NSO group (12.80 ± 0.91, 7.22 ± 1.45), and both were lower than the positive group of minoxidil (19.49 ± 2.23) (Figure 3a,b). Compared with the control group, the low NSO (10 μg/mL) decreased by 51.39%, while the high NSO (20 μg/mL) decreased by 72.58%. Compared with the minoxidil group, the low NSO (10 μg/mL) decreased by 34.33%, while the high NSO (20 μg/mL) decreased by 62.96%. Thus, these results indicate that NSO has a protective effect against H_2_O_2_-induced oxidative damage on DPCs.

### 2.3. The Effects of NSO on the Gene Expression Levels of DPC Proliferation and Antioxidation

A microarray analysis was performed on negative controls (H_2_O_2_-treated group) and 20 μg/mL NSO (NSO treatment group). Firstly, the difference in DEG between the two samples was checked. The comparison showed that 52.69% of genes were upregulated, and 47.31% of genes were downregulated (Figure 4a). The upregulation of antioxidant-related genes *TXNRD2*, *HMOX1*, ferroptosis-related gene *GPX4*, and dermal papilla cell growth-promoting genes *MAPK3*, *AKT2*, and *FGF*2 was observed after NSO treatment (Figure 4c,d). To demonstrate the effect of NSO on DPC, RT-PCR was used to validate the microarray findings. The data showed that the expression levels of *MAPK3*, *AKT2*, and *GPX4* in the NSO-treated group were 1.04-fold, 1.24-fold and 1.34-fold of those in the control group, respectively (Figure 4e); after 1.5 h of oxidative treatment (12 h of NSO pretreatment), the expression levels of *TXNRD2*, *FGF2*, and *HMOX1* in the NSO-treated group were 1.03-fold, 1.13-fold and 3.21-fold (Figure 4f); similarly, after 1.5 h of oxidative treatment (36 h of NSO pretreatment), the expression levels of *MAPK3*, *FGF2*, and *HMOX1* in the NSO-treated group were 1.36-fold, 1.77-fold and 1.39-fold of those in the control group, respectively (Figure 4g). These results suggest that NSO stimulates DPC growth and improves antioxidant capacity in the early stage of treatment.

### 2.4. Hair Growth-Promoting Effect of NSO in C57BL/6 Mice

To investigate whether topical NSO is beneficial for in vitro hair growth, NSO was applied to the depilated area of mice for two consecutive weeks. Each group had eight mice. In the first 7 days, it was observed that the skin on the back of the mice gradually turned from pink to grayish black, and in the second stage of NSO application (8–14 days), the hair of the mice began to grow significantly (Figure 5a). Both the experimental group and the positive group had significant hair growth-promoting effects compared to the control group (Figure 5b). The results indicated that the topical application of NSO can induce rapid hair growth in the early stages of resting mouse skin growth. To further examine the effect of NSO on the hair growth of the mice, the length and weight of hair and the number, depth, and dermal thickness of hair follicles were systematically compared. Compared with the control group, the 3% NSO group and 5% NSO group showed a significant enhancement in hair length and weight after 14 days of treatment (Figure 6a,b). As the duration of NSO therapy increases, the effect on hair growth improves (Figure 6c). Among them, the hair length of mice in the low-NSO (3%) and high-NSO (5%) treatment groups was 1.13 and 1.10 times longer than the positive group, while the hair weight was 1.53 and 1.45 times greater.

Histological analysis showed that the application of NSO can induce hair growth (Figure 7a). In the first week, the hair follicles in the low-NSO (3%) and high-NSO (5%) treatment groups were 63% and 57% more than those in the positive group. By the second week, it was 37% and 23% (Figure 7b,e). Compared with the first week, the hair follicle depth of each group increased significantly in the second week. In the first week, the hair follicle depth of the low-NSO (3%) and high-NSO (5%) treatment groups were 32% and 55% greater than that of the minoxidil group. The second week was 13% and 35% greater (Figure 7c,f). In the second week, most of the hair follicles in mice in the 3% minoxidil, 3% NSO, and 5% NSO groups were located deep in the subcutaneous tissue and had increased follicle density and elongated hair follicle cells, whereas most of the hair follicles in the control group were located in the epidermis and the number of hair follicles was low (Figure 7a). In terms of dermal thickness, both low NSO (3%) and high NSO (5%) were 36% and 19% thicker in the second week compared to the minoxidil group (Figure 7d,g). The data show that after NSO treatment, the number, depth, and thickness of hair follicles in mouse skin significantly increased, indicating that the topical use of NSO can stimulate hair growth.

### 2.5. GC/MS Profile of NSO Extracts

The GS/MS chromatogram of the NSO extract indicates the presence of 203 components, of which 5 have been quantified. Table 1 provides further explanation.

## 3. Discussion

This study found that arachidonic acid (AA) helps control hair growth and the hair cycle. Using in vivo and in vitro models, this study found that AA enhances hDPC viability and promotes the expression of several factors responsible for hair growth, in addition to promoting hair growth by inducing and prolonging anagen in resting C57BL/6 mice [35]. It was also found that treatment with polyunsaturated fatty acids, such as linoleic acid (LA), in vegetable oils derived from the seeds of *Malva verticillata* may alleviate testosterone-induced signaling molecules and induce the growth of HFDPCs by activating Wnt/*β*-catenin signaling [36]. Nutritional supplements with the main components of EPA and DHA have a positive effect on thinning hair [37]. In addition, research has shown that women who have hair loss may also have an increased risk of heart disease or metabolic disorders, and the early supplementation of polyunsaturated fatty acids like ω-3 is beneficial for the physical health of these patients [38]. A para rubber (Hevea brasiliensis (Willd. ex A. Juss.) Müll. Arg.)) seed oil with linoleic, palmitic, and stearic acids as its main components was shown to stimulate cell proliferation and produce cellular antioxidant activity in the human dermis with potency comparable to that of minoxidil, dutasteride, and vitamin C at the same tested concentrations [39]. 5α-reductase can convert testosterone into dihydrotestosterone. The main component of NSO, palmitic acid, can inhibit the activity of 5α-reductase, reducing the possibility of further transcription into androgenetic alopecia [39]. The results of GC-MS analysis showed that NSO contains substances that promote hair growth (EPA, palmitic acid, and linoleic acid), and palmitic acid, one of the main components of NSO, can inhibit the activity of 5α reductase and reduce the possibility of further transcription into androgenic alopecia, which may be effective in controlling hair loss.

To investigate the antioxidant capacity of NSO, we conducted cytotoxicity and antioxidant capacity tests on DPCs. The experimental results show that NSO has a similar promoting effect to minoxidil in DPC viability while effectively resisting oxidative damage.

Moreover, our study showed that NSO can regulate the pathway associated with hair growth development, as evidenced by the upregulation of MAPK, *FGF*, and mTOR after NSO treatment (Figure 4b). In HF, the MAPK protein is involved in regulating root tip growth, stimulating proliferation, and inducing and prolonging the growth period [40]. The mTOR pathway regulates cell proliferation and cycle development in the hair cycle by interacting with the MAPK pathway [41]. The interactions and communication in these pathways continue throughout the hair growth cycle. In our investigation, the NSO extract actively stimulated the DPC growth-related gene *FGF2*. *FGF2* sustains the high proliferation and multipotency of mesenchymal stem cells derived from human hair follicles, which is necessary for promoting the growth phase and HF morphogenesis [42]. Hydrogen peroxide can cause cell damage by producing ROS, which can downregulate cell growth, accelerate the cell aging process, and ultimately lead to hair loss [43]. In this study, we chose to treat DPCs with 200 μM H_2_O_2_ for 1.5 h, and then significant changes were observed in the cell survival rate and ROS production. *TXNRD2* eliminates free radicals by increasing the *SOD*, *CAT*, and *GSH* levels, thereby reducing ROS content [44]. In different types of vascular cells, *HMOX1* converts cellular heme into carbon monoxide (CO), biliverdin, and Fe^2+^, which is crucial in the fight against inflammation, oxidative damage, apoptosis, and thrombosis [45]. As a key cytoplasmic peroxidation inhibitor protein, the lack of *GPX4* enhances cellular lipid peroxidation. Iron accumulation, cytotoxic ROS, and lipid oxidation are characteristics of ferroptosis, and the inactivation of or reduction in *GPX4* will lead to ferroptosis [46]. Research has shown that ferroptosis inhibitors can protect cells from damaging cell stress or prevent cell death by increasing *GPX4* expression [47]. The inhibition and imbalance of genes associated with hair growth lead to an early entry into the regression period and a longer resting period, leading to hair loss eventually. The results show that NSO promotes cell proliferation and resists oxidative damage by regulating genes related to the proliferation (*FGF2*, *MAPK3*, and *AKT2*) and antioxidation (*TXNRD2*, *HMOX1*, *GPX4*) of DPC (Figure 4c and Figure 8).

## 4. Materials and Methods

### 4.1. Nannochloropsis salina Fermented Oil Extract Preparation

The *Nannochloropsis salina* fermented oil (NSO) was purchased from Innova Bay Technology Ltd. (Shenzhen, China). NSO was extracted as follows: cells were collected by centrifugation and then dried using a freeze dryer. The Soxhlet extraction method was used, whereby the freeze-dried cells were extracted with methanol in a Soxhlet device at 60 °C for 72 h. The extracted liquid was evaporated to dryness at 60 °C using a rotary evaporator [48]. NSO was dissolved in DMSO/EtOH solution with a volume ratio of 1:1 and a concentration of 50 mg/mL. The volume of the DMSO/EtOH solution was 0.2% of the volume of Duchenne Modified Eagle Medium (DMEM) [33].

#### GC-MS Analysis

The analysis of NSO’s composition followed the guidelines provided by earlier writers [49]. By mixing 500 mg of NSO with 10 mL of methanol solution containing 4% sulfuric acid and heating the reaction for one hour at 65 °C in a water bath, fatty acid methyl esters (FAMEs) were produced. The FAMEs were washed with 5 mL of deionized water and hexane and evaporated under a stream of nitrogen to give methylated fatty acids (MEFs). The MEFs from the previous step were dissolved in 2.5 mL of dichloromethane, and the MEFs were analyzed using gas chromatography/mass spectrometry (GC-MS, Agilent, Santa Clara, CA, USA). FAMEs were analyzed using an HP-5MS column (30.0 m × 250 μm × 0.25 μm, Agilent, Santa Clara, CA, USA). The split ratio was kept at 10:1, while the carrier gas was helium in constant-pressure mode. The column was heated from its starting temperature of 60 °C to 180 °C at a rate of 25 °C per minute. After that, the temperature was raised to 240°C for one minute at a rate of 3 °C min^−1^ and then to 250 °C for 5 °C min^−1^. The mass spectrometry library of the National Institute of Standards and Technology (NIST) was used to identify the fatty acids.

### 4.2. Cell Culture

The supplier of dermal papilla cells (DPCs) was Meisen Chinese Tissue Culture Collections. IHFDPC-SV40 T0500 special culture medium (obtained from Meisen Chinese Tissue Culture Collections, Hangzhou, China) was used to cultivate DPCs. The cells were maintained in a 37 °C culture incubator with 5% CO_2_ in it. Trypsin was added to the cells at a 1:2 ratio when the cell density became close to 80%.

#### 4.2.1. Cell Viability Assay

The impact of NSO on DPC viability was measured using the Cell Counting Kit-8 (CCK-8) test. As previously noted [33], DPCs were cultivated for twenty-four hours after being inoculated at a density of 5 x 10^3^ cells/well into a 96-well plate. DPCs were treated with NSO diluted to 7 different concentrations with culture medium (80, 40, 20, 10, 5, 2.5, 1.25 μg/mL). The CCK-8 assay was used to show cell viability. The plates were cultured for a time of 12 h, 36 h, and 60 h under 5% CO_2_ and 37 °C. Then, 10 μL of CCK-8 solution (HY-K0301, MCE, China) was added to each well, and the cells were incubated for 4 hours at 37 °C. The absorbance of the plate was measured at 450 nm using a microplate reader (Bio Tek Instrument Inc., Winooski, VT, USA). Independent biological duplicates were used in each experiment, which was run at least three times. The measured absorbance was averaged to determine cell proliferation, as shown below:

Cell proliferation(%) = [A2 − A0]/[A1 − A0] × 100%

A2:100 μL medium and 5 × 10^3^ cells mixed with NSO and 10 μL CCK-8 solution;

A1:100 μL medium and 5 × 10^3^ cells added 10 μL CCK-8 solution without NSO;

A0:100 μL medium mixed with 10 μL CCK-8 solution without cells.

#### 4.2.2. Antioxidant Activity of NSO

To detect the antioxidant effect of NSO on DPCs, hydrogen peroxide was used as the model of oxidation-induced damage. Antioxidant capacity was determined by reference to Bejaoui, M.’s description [50], and CCK-8 was used for detection. DPCs were inoculated onto a 96-well plate at a density of 8 × 10^3^ cells/well at 37 °C for 24 h. The cells were grown for 36 h in a medium with varying doses of NSO (as previously mentioned). After that, the cells were incubated for 1.5 h in a mixture with 0 μM and 200 μM H_2_O_2_. As mentioned above, CCK-8 was applied for monitoring the cells’ survival rate against H_2_O_2_-induced cytotoxicity.

#### 4.2.3. Determination of Active Oxygen Level

The redox condition of cells was found using the 2′-7′-Dichlorodihydrofluorescein diacetate technique (DCFH-DA) [51]. Using the reactive oxygen species fluorescence probe DCFH-DA (HY-D0940, MCE, Shanghai, China), the amount of reactive oxygen species (ROS) in DPCs was measured in order to determine the impact of NSO on that level [52]. Initially, as previously mentioned, the DPCs were cultured on 6-well plates at a density of 1 × 10^5^ cells per well in a medium containing different concentrations of NSO. After removing the previous medium, the cells were incubated for 1.5 h in a solution containing 0 μM and 200 μM H_2_O_2_. Subsequently, the old medium was removed, and each was thoroughly washed with PBS twice. Fresh medium was added, and we let it sit for two hours. Finally, 20 μM of DCFH-DA was added, and we let it incubate for half an hour. An inverted fluorescent microscope (Leica, Weitzlar, Germany) was used to take pictures of each well after it had been cleaned twice with PBS to remove any remaining probe. Image J V. 2.9.0 (National Institutes of Health, Bethesda, MD, USA) was used to analyze the average fluorescence intensity of each group.

### 4.3. Quantitative Real-Time Polymerase Chain Reaction (qRT-PCR)

The total RNA was extracted and collected from both experimental and control DPCs using a Trizol reagent. A NanoDrop2000 spectrophotometer (Thermo Scientific, Waltham, MA, USA) was applied to analyze the concentration and purity (A260/280 ratio > 2.0) of the RNA samples. The Prime Script™ RT kit and gDNA eraser (Takara Biotechnology) were used for the reverse transcription synthesis of cDNA. cDNA was amplified using specific primers through the TB Green Premix Ex Taq™ II kit (Takara Biotechnology). The sequences of the primers utilized are displayed in Table 2. ABI Quant Studio 6 Flex (Applied Biosystems, Foster, CA, USA) was used to conduct real-time PCR experiments. As directed by the manufacturer, there should be one cycle at 95 °C for 30 s, followed by 40 cycles at 95 °C for 5 s, and finally one cycle at 60 °C for 30 s. β-actin, the “housekeeping gene,” was used to conduct normalization. As previously mentioned, the 2^−∆∆ct^ technique was employed to demonstrate variations in gene expression [51].

### 4.4. Experimental Animals

Prior to the start of the experiment, male C57BL/6 mice, aged six weeks (TOP Biotechnology Co., Ltd., Shenzhen, China), were given unrestricted access to food and water at a temperature of 22 ± 2 °C, a relative humidity of 50 ± 5%, and a 12 h light/dark cycle. This allowed the mice to acclimate to their new environment for a week. Six-week-old C57BL/6 mice had their back area (2 cm × 4 cm) shaved with animal scissors. After shaving, the hair follicles of mice were all in a resting phase and had a pink appearance. A total of 100 μL of 3% H_2_O_2_ was topically applied on the depilated area of the mice (2 cm × 4 cm) for 30 min as an oxidative stimulus. The area was then rinsed with tap water at 25 °C [10]. Forty mice (*n* = 8/group) were randomly divided into 5 groups based on different topical applications: blank control group (control), solvent group (ethanol/DMSO = 1:1), positive control group (3% minoxidil), and experimental group (3% and 5% NSO, diluted with ethanol and DMSO). Each compound (100 μL) was topically applied to the shaved back skin once a day, 7 days a week, for 2 consecutive weeks. Approved by the Institutional Animal Care and Use Committee of Shenzhen University Medical School (IACUC-202300118), all animal studies were carried out in compliance with the National Institutes of Health’s Guidelines for the Care and Use of Experimental Animals.

#### 4.4.1. Hair Growth Observation

To evaluate the hair growth of each group, photos of the animals were taken in the first and second weeks after the start of topical application. Due to the time-synchronous hair growth cycle of the C57BL/6 mouse strain, the hair growth-promoting activity of NSO was studied by comparing the areas of trimmed back skin treated with NSO that changed from pink (in the stage of telogen) to gray (in the stage of anagen). Differences were compared by calculating the proportion of the completely black area to the entire back skin using Image J V. 2.9.0 (National Institutes of Health, Bethesda, MD, USA) [53].

#### 4.4.2. Histological Analysis

After the euthanasia of mice, the skin on the back of the mice was separated and fixed in 10% neutral formalin buffer, then embedded in paraffin and sectioned at a thickness of 5 μm, then stained with hematoxylin and eosin (H&E) [54].

#### 4.4.3. Determination of Hair Length

Hair was randomly removed from previously hairless areas in a group of mice. The average length of 20 stochastically selected hairs was measured, and the result was recorded as the average length of 20 hairs ± standard deviation (SD). The length was measured after 7 and 14 days of treatment [55].

#### 4.4.4. Determination of Hair Weight

The hair weight was measured at the end of 14 days of treatment. The neonatal hair on the black area of the back of each mouse was carefully clipped and collected using animal scissors, and the weight of the shaved hair in each group of mice was determined using an analytical balance.

#### 4.4.5. Determination of Hair Follicle Length and Dermal Thickness

The response of mice hair follicles to the NSO extract was observed 14 days after topical application. Photographs were taken using a light microscope. Photographs obtained using HE staining (magnification 100×) were used to determine the hair follicle numbers, follicle depth, and dermal thickness. The captured images were analyzed using Image Viewer (DPVIEW V2.0.4.0422, Shenzhen Shengqiang Technology Co, Ltd., Shenzhen, China) The outcomes were recorded as the mean ± standard deviation for each treatment (*n* = 3).

### 4.5. Statistical Analysis

All the experimental data were expressed as the mean ± standard deviation (S.D.) of at least three independent experiments. Statistical analyses were performed using a one-way analysis of variance ANOVA, and t-tests were used to determine the statistical significance of differences. Statistical significance was defined as *p* < 0.05, with an asterisk (*) indicating the significance of the difference, * *p* < 0.05, ** *p* < 0.01, *** *p* < 0.001, and **** *p* < 0.0001. Graphs were made using GraphPad Prism 9.0.0 (San Diego, CA, USA).

## 5. Conclusions

NSO, mainly composed of EPA, palmitic acid, and linoleic acid, is safe for human dermal papilla cells. This oil can protect cells from oxidative damage and has cell proliferation activity comparable to minoxidil. The positive effect of NSO on cell proliferation (increased up to 4.30 times) was observed since the initiation stage (12 h). We also found that the longer the cultivation time of NSO, the better its resistance to oxidative stress. When the cultivation time was 60 h, the relative survival rate was 8.99 times that of the positive group. We further validated the effect of NSO on DPC growth through the RT-PCR validation of the microarray analysis. In particular, the expression level of genes related to antioxidants (*HMOX1*) was 3.21 times greater than that of the control group, whereas the expression level of genes that promote proliferation (*FGF2*) was 1.77 times higher. For two weeks in a row, mice were given topical NSO in the hair removal area to test its potential for promoting in vitro hair growth. C57BL/6 mice treated topically with NSO for 2 weeks showed a much greater increase in the hair growth index and hair growth area than mice treated with minoxidil. These results suggest that NSO stimulates the growth of DPCs and improves their antioxidant capacity during the early stages of treatment. NSO induced the anagen phase, suggesting that NSO could be used as a topical product to improve hair growth or inhibit hair loss. However, further research is needed to investigate the hair growth-promoting effects of NSO, such as confirming these findings in human clinical trials, to ensure the efficacy of NSO as a safe and effective natural anti-hair loss substance.

## Figures and Tables

**Figure 1 ijms-25-08231-f001:**
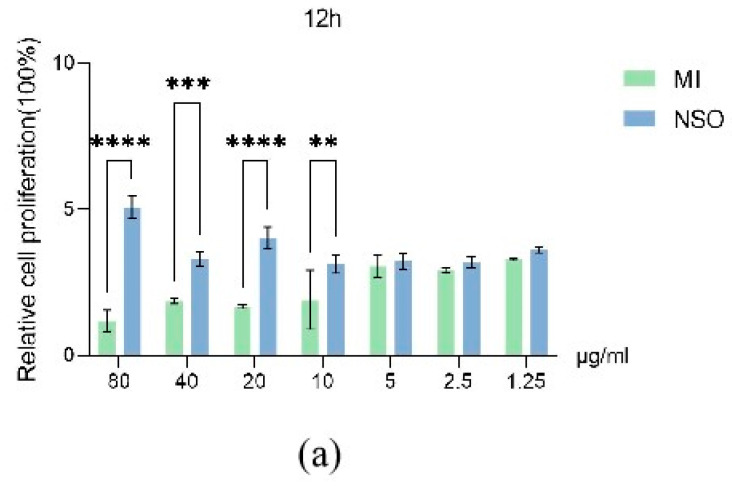
The effect of NSO on the cell viability of DPCs. The cck-8 method was used to determine the cell proliferation of NSO cultured at different concentrations (80, 40, 20, 10, 5, 2.5, 1.25 μg/mL) for 12 h (**a**), 36 h (**b**), and 60 h (**c**), compared to the positive control minoxidil. The value is significantly different from the control group and the treatment group (* *p* < 0.05, ** *p* < 0.01, *** *p* < 0.001, **** *p* < 0.0001). MI: minoxidil; NSO: *Nannochloropsis salina* fermented oil.

**Figure 2 ijms-25-08231-f002:**
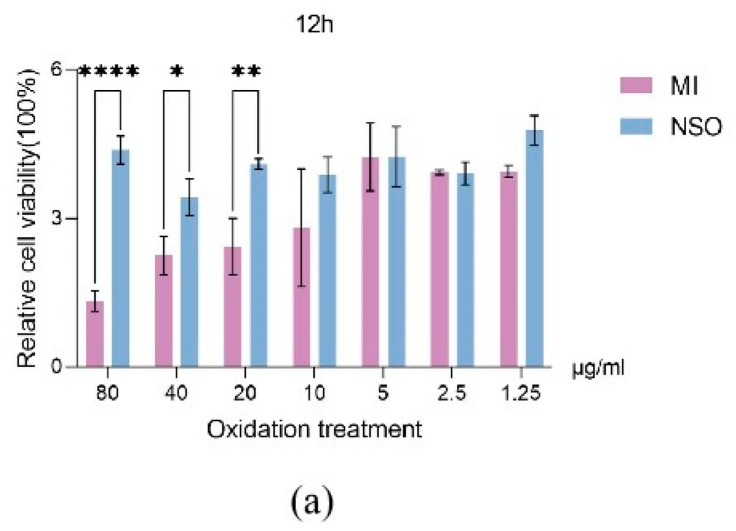
The detection of the effect of NSO on H_2_O_2_-induced cytotoxicity in cells. Different concentrations of NSO (80, 40, 20, 10, 5, 2.5, 1.25 μg/mL) were applied to cells for 12 h (**a**), 36 h (**b**), and 60 h (**c**) and then treated with 200 μM H_2_O_2_ for 1.5 h. Cell viability was measured using the CCK-8 method. The value is significantly different from the control group and the treatment group (* *p* < 0.05, ** *p* < 0.01, **** *p* < 0.0001). MI: minoxidil; NSO: *Nannochloropsis salina* fermented oil.

**Figure 3 ijms-25-08231-f003:**
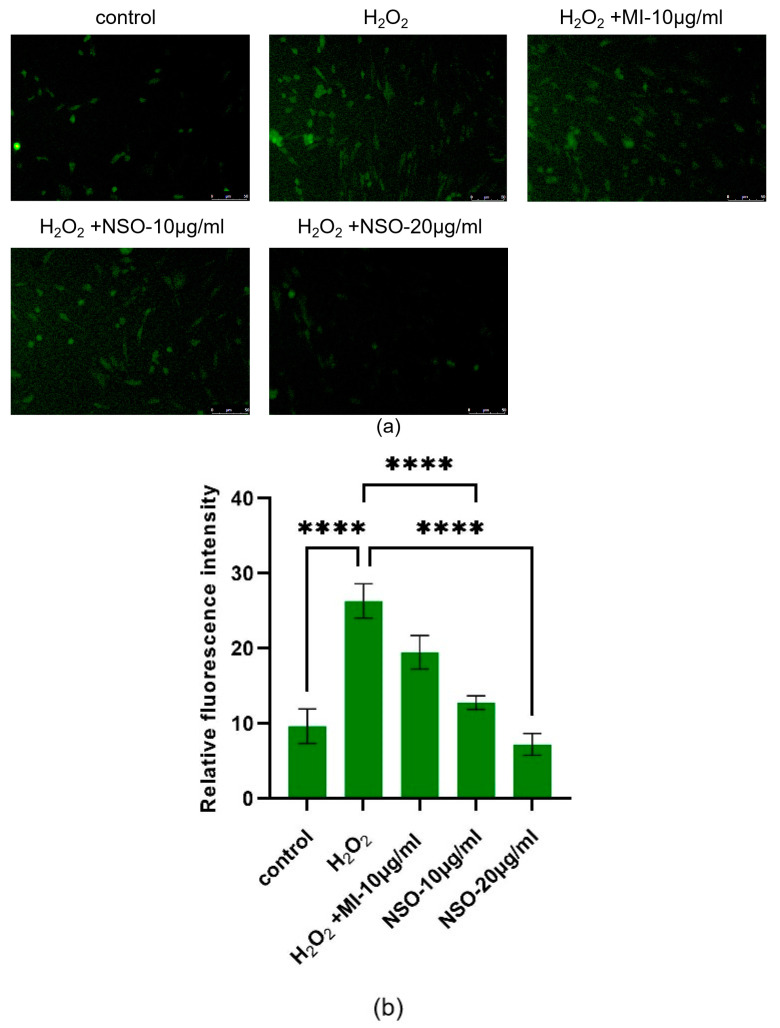
(**a**) The reactive oxygen species fluorescence probe DCFH-DA was used to detect the level of reactive oxygen species in DPCs (The scale bar = 50 μm). (**b**) The average fluorescence intensity of each group was measured using Image J software V. 2.9.0. The difference between the control group and the treatment group was statistically significant (**** *p* < 0.0001). MI: minoxidil; NSO: *Nannochloropsis salina* fermented oil.

**Figure 4 ijms-25-08231-f004:**
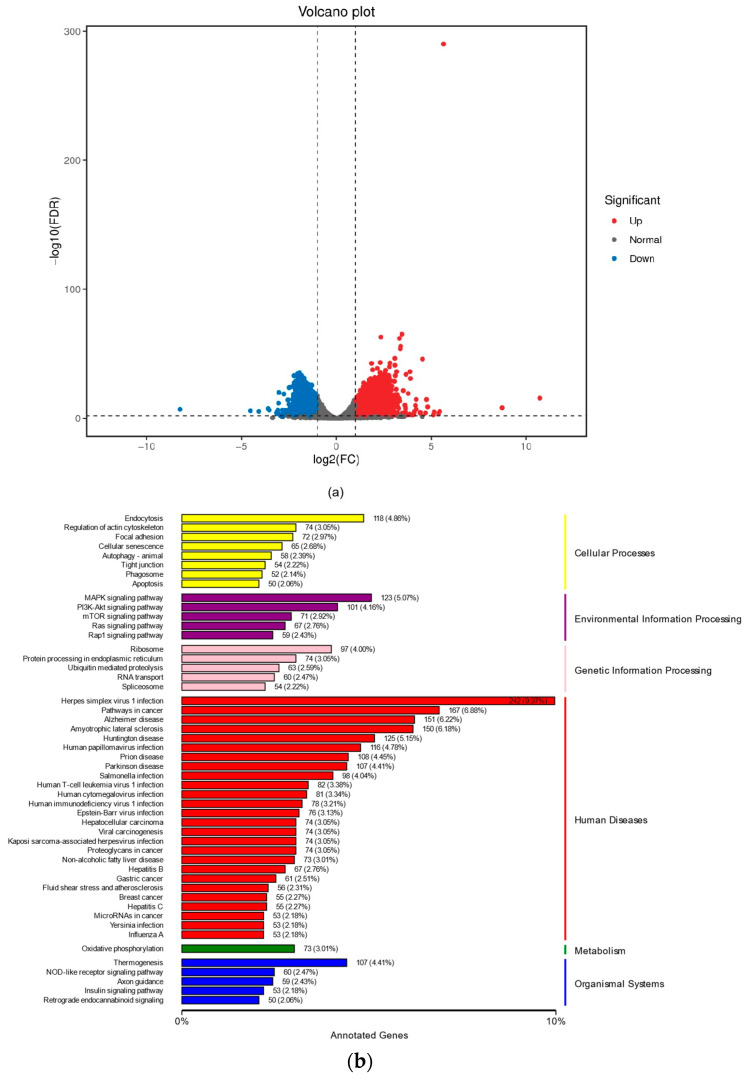
Microarray analysis comparison of gene expression pattern in DPCs treated with NSO. (**a**) Volcano plot showing DEGs. X-axis represents fold change, and Y-axis represents *p*-value (log 10). Red color represents upregulated DEGs, blue color represents downregulated DEGs, and gray color represents nonregulated genes. (**b**) Significantly enriched KEGG pathways by upregulated DEGs (analyzed using DAVID and GSEA) in NSO extracts. Each bar is arranged according to significance (*p*-values) and represents number of DEGs. (**c**) Heatmap of differentially expressed genes of control and CDN groups. (**d**) Heatmap of differentially expressed genes of CDH and CDNH groups. (**e**) Gene expression assay of antioxidant and hair growth markers in control and CDN groups: *MAPK3*, *AKT2*, and *GPX4*. (**f**) Gene expression assay of antioxidant and hair growth markers in CDH and CDNH groups (NSO treatment for 12 h): *TXNRD2*, *FGF2*, and *HMOX1*. (**g**) Gene expression assay of antioxidant and hair growth markers in CDH and CDNH groups (NSO treatment for 36 h): *MAPK3*, *FGF2*, and *HMOX1*. Significant difference from respective control (* *p* < 0.05, ** *p* < 0.01, *** *p* < 0.001). Control: cells without any treatment. CDH: cells treated with medium containing 200 µM H_2_O_2_, D stands for DMEM; CDN: cells treated with 20 µg/mL NSO; CDNH: cells treated with 20 µg/mL NSO and 200 µM H_2_O_2_. *MAPK3*: mitogen-activated protein kinase 3; *AKT2*: serine/threonine kinase 2; *GPX4*: glutathione peroxidase 4; *TXNRD2*: thioredoxin reductase 2; *FGF2*: fibroblast growth factor 2; *HMOX1*: heme oxygenase 1; *MAPK3*: mitogen-activated protein kinase 3.

**Figure 5 ijms-25-08231-f005:**
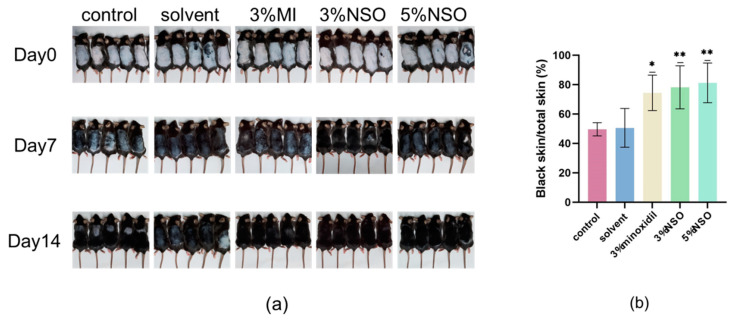
A comparison of hair growth effects in C57BL/6 mice. We applied 3% and 5% NSO to the depilated area of mice for 14 consecutive days and evaluated the hair growth of mice after 2 weeks. (**a**) Mice treated with solvent (DMSO/ethanol = 1:1), 3% minoxidil, 3% NSO, and 5% NSO were compared with the control group within two weeks. (**b**) Compared with the control group, the group treated with 5% NSO had a significant promoting effect on mice hair growth (* *p* < 0.05, ** *p* < 0.01). The Image J program was used to measure the size of the black skin area occupying the entire hair removal area. The data are represented as the mean ± SD (*n* = 8). 3% MI: 3% minoxidil; 3% NSO: 3% *Nannochloropsis salina* fermented oil; 5% NSO: *Nannochloropsis salina* fermented oil.

**Figure 6 ijms-25-08231-f006:**
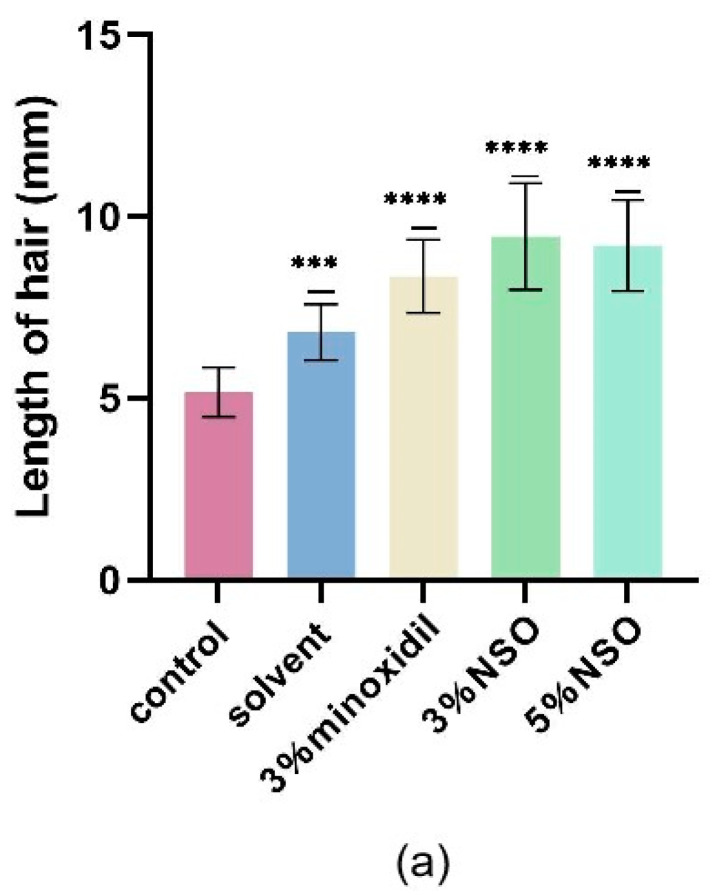
(**a**) The length of the mouse hair was measured on day 14. (**b**) The mouse hair weight was measured on day 14. (**c**) The growth rate of mouse hair. The length of mice hair was measured on days 5, 10, and 15. The difference between the control group and the treatment group was statistically significant ( *** *p* < 0.001, **** *p* < 0.0001). 3% NSO: 3% *Nannochloropsis salina* fermented oil; 5% NSO: *Nannochloropsis salina* fermented oil.

**Figure 7 ijms-25-08231-f007:**
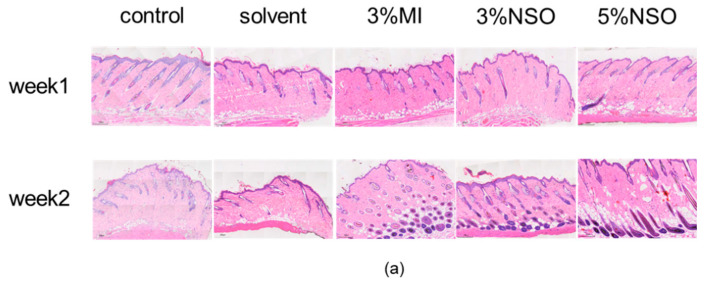
(**a**) A representative microscopic cross-section of HE-stained skin tissue (Magnification: 100×). Changes in the hair follicle number, hair follicle depth, and dermal thickness in C57BL/6 mice. (**b**,**e**) The number of hair follicles on the back skin of mice was measured in the first and second weeks. (**c**,**f**) The hair follicle depth of the mouse back skin was measured in the first and second weeks. (**d**,**g**) The dermal thickness of the mouse back skin was measured in the first and second weeks. Determined by the magnification of the HE-staining section at 50× for observation. Data represented as the mean ± SD (*n* = 3). Significant difference from respective control (* *p* < 0.05, ** *p* < 0.01, *** *p* < 0.001, **** *p* < 0.0001). 3% MI: 3% minoxidil; 3% NSO: 3% *Nannochloropsis salina* fermented oil; 5% NSO: 5% *Nannochloropsis salina* fermented oil.

**Figure 8 ijms-25-08231-f008:**
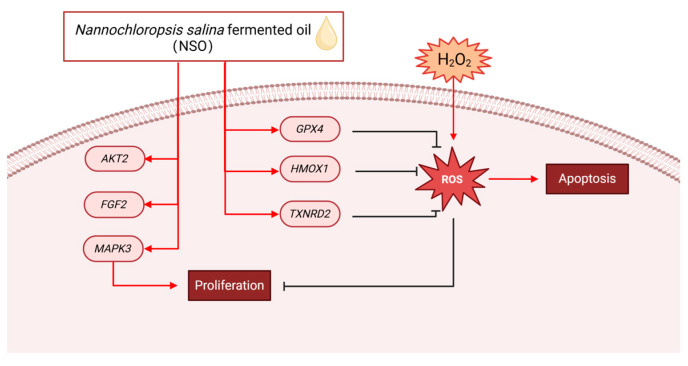
NSO reduces the accumulation of ROS caused by hydrogen peroxide by increasing the expression levels of *GPX4*, *HMOX1*, and *TXNRD2* and promotes the proliferation of dermal papilla cells by increasing the expression levels of *AKT2*, *MAPK3*, and the DPC growth-associated factor *FGF2*. *AKT2*: serine/threonine kinase 2; *MAPK3*: mitogen-activated protein kinase 3; *FGF2*: fibroblast growth factor 2; *GPX4*: glutathione peroxidase 4; *HMOX1*: heme oxygenase 1; *TXNRD2*: thioredoxin reductase 2. Created with BioRender.com.

**Table 1 ijms-25-08231-t001:** Five components with higher content in NSO extract.

EPA	(Z)-9-PA	LA	PA	MA
23.25% ± 2.54%	26.01% ± 0.50%	15.33% ± 1.39%	9.92% ± 1.40%	5.70% ± 0.50%

EPA, eicosapentaenoic acid; PA, palmitic acid; LA, linoleic acid; MA, myristic acid.

**Table 2 ijms-25-08231-t002:** Primer sequence used for real-time PCR analysis.

Gene	Primer
*β-actin*	F: 5′-GTGGAGTCATACTGGAACATGTAG-3′ R: 5′-AATGGTGAAGGTCGGTGTG-3′
*TXNRD2*	F: 5′-GCGAGTTCCAGAAACCAGGA-3′ R: 5′- AATATGCATGGTAAACATTGCTGT-3′
*HMOX1*	F: 5′-CTTCTTCACCTTCCCCAACA-3′ R: 5′-AGCTCCTGCAACTCCTCAAA-3′
*GPX4*	F: 5′-CAAAGTCCTAGGAAACGCCC-3′ R: 5′-CCTTGGCTGAGAA TTCGTGC-3′
*MAPK3*	F: 5′-TGCTGGACCGGATGTTAACC-3′ R: 5′-CTCATCCGTCGGGTCATAGT-3′
*AKT2*	F: 5′-TACATCAAGACCTGGAGGCC-3′ R: 5′-GGGGGTAGAGTCTGATCAGG-3′
*FGF2*	F: 5′-TCAAGCTACAACTTCAAGCAGA-3′ R: 5′-AGCCAGTAATCTTCCATCTTCCT-3′

## Data Availability

Data is contained within the article.

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
