# Peer review of "Effects of Nannochloropsis salina Fermented Oil on Proliferation of Human Dermal Papilla Cells and Hair Growth"

_ijms, 2024, doi:10.3390/ijms25158231_

Round 1

Reviewer 1 Report

Comments and Suggestions for Authors

General comments:

The paper has a very well written introduction. I also have no major comments on the selection of methodologies and their description. The literature cited also seems to be sufficient for sections 1-3. The manuscript is generally well written and contains many important research findings from a treatment perspective of hair loss. However, section 4 - Discussion - needs to be completed. Detailed comments are included below.

 1.      A very well-written introduction, but at the end of the section (p. 3, lines 110-113 instead of a description of what is in the manuscript: “This study discussed the effect of the extract of Nannochloropsis salina fermented oil on the hair growth of model mice, evaluated its effect on promoting hair growth, and clarified its mechanism, providing a valuable theoretical reference for the further research of Nannochloropsis salina fermented oil in the prevention and treatment of hair loss.” Authors should include the purpose/aim of the work (“the aim of the work was to…..”).

2.      Section 2.1 please add the concentration of NSO used in DMSO/EtOH solution.

3.      P. 3 line 119: What is the “DMEM medium” (write it with no acronyms, it is the first use of the term).

4.      P.3 line 140 there is “CO2”, there should be “CO2”.

5.      P. 3 line 143: “CCK-8 test” (write it with no abrreviations, it is the first use of the term).

6.      P.6 line 254: “MI” (write it with no acronyms, it is the first use of the term).

7.      P. 6 -7 Figure 1-2:  I recommend that drawings a, b and c to be placed one under the other (enlarged) or attached as an annex. They are small and not very clear.

8.      P. 7 Figure 3: I recommend that drawings a to be placed one under the other (enlarged). They are very important in this work.

9.      P. 8 Figures 4a and 4b:  These drawings are crucial to the work and should be enlarged.

10.  P. 9 line 340: not “has” but “had”.

11.  P. 10 Figure 6: I recommend that drawings a to be placed one under the other (enlarged).

12.  P. 11 Figure 7b-g: I recommend that drawings a to be placed one under the other (enlarged) or attached as an annex. They are small and not very clear.

13.  Section 4 lacks a comparison of the results obtained in this work with those obtained by other researchers. I understand that the authors are the first to study effects of Nannochloropsis salina fermented oil on proliferation of human dermal papilla cells and hair growth, but reference should be made to the results obtained by other research teams. Otherwise it is difficult to know whether the results obtained in the paper are significant in the field. Even if other teams have studied the effects of other chemicals, extracts or oils, they should be cited or it should be noted that these methodology were not used in this way and there is nothing to compare the results with. This is my main objection to the paper, but the authors could complete the paper in several sentences.

14.  P. 13 line 458 Section 5 authors wrote: “These data all indicate that the NSO effectively promotes hair growth.”- in order to conclude this you have to compare the results with something - not only with the control sample but with other researchers. See note above.

Reviewer 2 Report

Comments and Suggestions for Authors

1. (Material&Method) Line 116: I recommend explaining Nannochloropsis salina fermented oil (NSO) in more detail and attaching a reference.

2. Fig3: It would be better to modify the description in the photo (a) and the x-axis label in (b). For example, damage --> H2O2 ; Minoxidil --> H2O2 + MI-10ug/ml etc.

3. Fig. 4: It would be good to increase the resolution of panel (b), and please write an explanation of the abbreviations C, DH, and CNH shown in (c) and (d) in figure legendary.

4. Fig. 5 (a): In the day 14 photo of the solvent-treated group, the skin of the 5th mouse is very different from the skin of the other 4 mice. Please explain this.

5. Fig. 8: It would be better to specify MAPK3 and AKT2 together in the figure.
